# Nongenomic Activities of Vitamin D

**DOI:** 10.3390/nu14235104

**Published:** 2022-12-01

**Authors:** Michał A. Żmijewski

**Affiliations:** Department of Histology, Faculty of Medicine, Medical University of Gdańsk, PL-80211 Gdańsk, Poland; mzmijewski@gumed.edu.pl; Tel.: +48-58-3491455

**Keywords:** vitamin D, VDR, nongenomic response, PDIA3, ultraviolet radiation, megalin, mitochondria, vitamin D analogs

## Abstract

Vitamin D shows a variety of pleiotropic activities which cannot be fully explained by the stimulation of classic pathway- and vitamin D receptor (VDR)-dependent transcriptional modulation. Thus, existence of rapid and nongenomic responses to vitamin D was suggested. An active form of vitamin D (calcitriol, 1,25(OH)_2_D_3_) is an essential regulator of calcium–phosphate homeostasis, and this process is tightly regulated by VDR genomic activity. However, it seems that early in evolution, the production of secosteroids (vitamin-D-like steroids) and their subsequent photodegradation served as a protective mechanism against ultraviolet radiation and oxidative stress. Consequently, direct cell-protective activities of vitamin D were proven. Furthermore, calcitriol triggers rapid calcium influx through epithelia and its uptake by a variety of cells. Subsequently, protein disulfide-isomerase A3 (PDIA3) was described as a membrane vitamin D receptor responsible for rapid nongenomic responses. Vitamin D was also found to stimulate a release of secondary massagers and modulate several intracellular processes—including cell cycle, proliferation, or immune responses—through wingless (WNT), sonic hedgehog (SSH), STAT1-3, or NF-kappaB pathways. Megalin and its coreceptor, cubilin, facilitate the import of vitamin D complex with vitamin-D-binding protein (DBP), and its involvement in rapid membrane responses was suggested. Vitamin D also directly and indirectly influences mitochondrial function, including fusion–fission, energy production, mitochondrial membrane potential, activity of ion channels, and apoptosis. Although mechanisms of the nongenomic responses to vitamin D are still not fully understood, in this review, their impact on physiology, pathology, and potential clinical applications will be discussed.

## 1. Vitamin D—Overview

Vitamin D is present in almost forms of life, from phytoplankton to humans, and is considered one of the oldest hormones on Earth [1,2,3]. In humans, the classic endocrine function of this powerful secosteroid is regulation of calcium–phosphorus homeostasis in order to maintain proper function of the body [4]. Initially, vitamin D has been described as a vitamin because it can be acquired from the diet, mainly from fatty fish, fish oil, milk products, and some mushrooms. However, for humans, the major and a natural source of vitamin D is its photochemical production by 7-dehydrocholesterol (7-DHC) [5]. This process is stimulated by ultraviolet type B (UVB) radiation (280–320 nm) and takes place in the basal layer of epidermis of the skin [6,7]. To be exact, photodegradation of the B-ring of 7-DHC resulted in production of three isomers: previtamin D, tachysterol, and lumisterol. Those byproducts are subjected to time-dependent thermal conversion to vitamin D, which—after release from the cells—enters the circulatory system and is transported to all organs by vitamin-D-binding protein (DBP) [8,9,10]. Regardless of the source, vitamin D requires two subsequent hydroxylations to achievement its full hormonal activity [3,5,11]. First, 25-hydroxylase (CYP2R1) facilitates the production of 25-hydroxyvitamin D_3_ (25(OH)D_3_) in the liver`s hepatocytes. This is the major metabolite of vitamin D, and its serum level is widely used as an indicator of vitamin D status [3,12,13,14,15]. The second hydroxylation takes place in proximal tubules of the kidney and requires activity of 1α-hydroxylase (CYP27B1). The final product is a fully active hormone—1,25(OH)_2_D_3_, calcitriol [16,17]. Serum levels of vitamin D are tightly regulated on two levels. Firstly, an excess of sun (ultraviolet radiation type B) leads to photodegradation of vitamin D and the formation of suprasterols [18]. Secondly, elevated levels of 1,25(OH)_2_D_3_ stimulate the expression of CYP24A1, which is 24-hydroxylase and is responsible for the catabolism (deactivation) of both 25(OH)D_3_ and 1,25(OH)_2_D_3_ [6]. Several other metabolites of vitamin D have been described in recent years, including 3-epi analogs [19,20,21], lactones [22], and CYP1A1 (CYP450scc) metabolites [23,24,25,26]. However, their biological function still is far from being understood.

1,25(OH)_2_D_3_ regulates a variety of cellular processes thought activation of a nuclear receptor—VDR. It binds to VDR, which forms a complex with its coreceptor RXR and is subsequently translocated into the nucleus. Upon activation, the VDR-RXR complex regulates expression from hundreds to more than 3000 genes in the human genome depending on cell type and physiological conditions [27,28,29]. Interestingly, recent studies have shown that not only 1,25(OH)_2_D_3_ can affect the expression of genes. 25(OH)D_3_ also binds to VDR but with an affinity approximately 1000 times lower compared to 1,25(OH)_2_D_3_. Transcriptomic analyses of human peripheral blood mononuclear cells (PBMCs) revealed similar patterns of genes affected by 25(OH)D_3_ treatment at 1000 or 10,000 nM concentrations compared to 10 nM 1,25(OH)_2_D_3_ [30,31]. Similar overlapping patterns of expression were also observed in the prostate cancer cell line LNCaP [32]. Intestinally, the metallothionein 2A gene was identified as a unique target for 1,25(OH)_2_D_3_ but not for 25(OH)D_3_ [33]. Other nonclassic metabolites of vitamin D, such as 20(OH)D_3_ and 20,23(OH)_2_D_3_, not only interact with VDR but also target other transcription factors, including RORα and RORγ [34,35] or AhR [28]. These experiments led to the detection of unique genes regulated by alternative nuclear receptors. Furthermore, it seems that the genes coding transcription factors (TFs) are activated by 1,25(OH)_2_D_3_, which in turn regulates the expression of additional sets of genes [36], which could be described as secondary genomic response. Recently, regulatory effects of 1,25(OH)_2_D_3_ on the expression of microRNAs and long noncoding RNAs have been uncovered as a potential anticancer mechanism [37,38].

The main target of vitamin D is the regulation of the calcium–phosphate homeostasis, which contributes to maintaining proper bone health [1,11,13,39,40,41]. Furthermore, the presence of VDR is already well documented in the cells of various organs, and it was established that proper function of musculoskeletal, immune, nervous, and cardiovascular systems as well as regeneration of epithelial barriers strongly depends on vitamin D. Vitamin D deficiency (25(OH)D_3_ level below 20 ng/mL) was not only described as a risk factor for rickets or osteoporosis [39,42] but also impairs the function of the immune system [43,44,45,46,47] and its response to pathogens, including influenza viruses [48] and coronaviruses (SARS-CoV-2) [49]. Vitamin D supplementation resulting in adequate level of vitamin D (preferably 30 ng/mL of 25(OH)D_3_ in the serum) improves muscle performance and reduces falls in vitamin-D-deficient older adults [50]. It also has a neuroprotective role [51,52] and prevents and reduces outcomes of psoriasis [53]. Furthermore, it was suggested that even higher levels of vitamin D (40–60 ng/mL of 25(OH)D_3_ in the serum) are beneficial, reducing the occurrence and severity of multiple types of cancer [23,54,55,56,57,58,59,60] and displaying neuroprotective [61] and cardioprotective properties [62,63]. Consequently, high doses of vitamin D were found to be beneficial in the prevention or treatment of several diseases, including cancer [23,54,56,58,59,64], neurodegenerative disorders, autoimmune diseases [43,45,65,66], psoriasis [67,68], and preeclampsia [69]. This must be of special interest because vitamin D deficiency is still a global problem [21,39,55,70,71,72]. Furthermore, seasonal changes in vitamin D levels that were observed pointed to the necessity of its proper supplementation, especially in the winter season [12,13,72].

In spite of the well-established role of VDR in the biological activity of vitamin D, not all effects of this powerful hormone could be liked to VDR-driven regulation of the expression of the genes. Firstly, some proven physiological effects can be observed within seconds or minutes after stimulation with vitamin D. These do not include activation of gene expression and subsequent protein synthesis, which take at least a few. Thus, the so-called rapid nongenomic responses to vitamin D have been described. The presence of membrane receptor for vitamin D has been suggested as an analogy to other steroid hormones [73]. Interestingly, in spite of the fact that VDR is a nuclear receptor, its involvement was not fully excluded. However, rapid responses do not rely on its transcriptional activity [74]. In this review, I will discuss different aspects of alternatives to the genomic activities of vitamin D. Figure 1 visualizes our current understanding of the variety of cellular processes activated by vitamin D and its metabolites.

## 2. Direct Effects of Vitamin D on Calcium Transport

The idea of alternative pathways activated by vitamin D arose from an observation of rapid (1–10 min) influx of calcium induced by calcitriol in osteogenic sarcoma cell line ROS 17/2.8 [75] or myocytes isolated from chicken embryonic heart [76]. Consequently, transcaltachia was described as a rapid—measured in seconds to minutes—resorption of calcium by enterocytes in response to calcitriol [77]. Interestingly, the rapid influx of calcium requires as low as picomolar concentrations of calcitriol (0.13 nM [78] or 0.3 nM [79]). Later, activation of rapid nongenomic responses to calcitriol was confirmed in vivo in chicken [80], in cultured mice chondrocytes with VDR knockout, and in osteoblastic cell line ROS 24/1 deprived of VDR expression [81]. The time frame (seconds to minutes) of observed calcium transport excludes involvement of the genomic pathway of response to vitamin D. Thus, the idea of membrane response was suggested.

## 3. Transmembrane Transport of Vitamin D

It is still presumed, according to free hormone hypothesis, that vitamin D can freely diffuse through the membranes and does not require transporter or channel to enter the cells [82], as can other lipophilic steroids. However, only ~0.04% of 1,25(OH)_2_D_3_ is considered to be a free hormone in circulation. Furthermore, the major circulating metabolite of vitamin D–25(OH)D_3_ is mainly carried by vitamin-D-binding protein (DBP; ~85%) and albumin (~15%). Thus, only a fraction of 25(OH)D_3_ is free (~0.03%) and can be directly absorbed by the target cells. In an addition, the affinity of 1,25(OH)_2_D_3_ to DBP is 10 to 100 times lower in comparison to 25(OH)D_3_ but is sufficient to be its main transporter [8]. This also raises an important question: How do vitamin D metabolites bound with high affinity to DBP become freed and enter into cells? It is well established that other hormones, including steroids and thyroid hormones are bound with corresponding globulins in circulation and require carrier proteins for efficient cellular uptake of their complexes [83]. Surprisingly, early studies revealed that deletion of DBP did not result in vitamin D deficiency under normal vitamin D supplementation, which strongly supported the free hormone hypothesis [84]. On the other hand, a deletion of megalin (LRP2) 600-kDa transmembrane glycoprotein, which functions as an endocytic receptor, results in vitamin D deficiency. Later studies reviled that megalin and two additional proteins—cubilin (CUBN) and disabled-2 (DAB2)—are required for effective endocytosis of DBP/vitamin D complexes. It was shown that deletion or deactivation of any of those proteins results in vitamin D deficiency, hypocalcaemia, and subsequent deterioration of bone structure and function [85,86,87].

Although expression of megalin, as well as megalin-mediated endocytosis of DBP protein, was shown in several tissues, including muscle [88], mammary gland [89], placenta, prostate, colon epithelium [90], and skin tissues, the best-studied model is epithelium of the proximal convoluted tubules of the kidney [91]. It was shown that the megalin complex is responsible for rapid and efficient resorption of small proteins, including DBP binding vitamin D, and this mechanism is necessary to maintain an optimal level of vitamin D in circulation. Interestingly, epithelial cells of proximal convoluted tubules express CYP27B1 and CYP24A1. Thus, the kidney is the primary location of both activation and deactivation of vitamin D [85,86,92]. Recent studies also postulated the role of megalin in membrane to mitochondria trafficking of various proteins (angiotensin II, stanniocalcin-1, and TGF-β) [93,94], which raises the question of whether the same mechanism could be utilized by vitamin D in complex with DBP. In summary, the role of the megalin-mediated transport of DBP-bound vitamin D metabolite in a rapid membrane response to this secosteroid and its mitochondrial targeting is currently under discussion and intensive investigation.

## 4. Membrane-Bound Receptor(s) and Targets for Vitamin D

As opposed to its genomic counterparts, nongenomic vitamin D responses appear rapidly (within a range of seconds or minutes) and are not susceptible to actinomycin D or cycloheximide. Therefore, they do not require expression of genes or protein synthesis [95]. Thus, a direct interaction of this secosteroid with membrane-bound or intracellular macromolecules was suggested as an acceptable explanation [96,97]. In fact, several steroids and other small regulatory molecules bind to membrane receptors in addition to their nuclear receptors, activating rapid responses [73,97]. Thus, in an analogy, an involvement of a G-coupled membrane receptor (GPCR) in vitamin D signaling was suggested, and its presence initially confirmed in membranous cell extracts [76,98]. Furthermore, the existence of membrane-associated activities of vitamin D was supported by observation of a rapid and efficient mobilization of the secondary messengers, including cAMP and calcium, triggered by 1,25(OH)_2_D_3_ [75,79]. Other studies documented fast (15–300 s) activation of phospholipase C (PLC) and calcium influx in response to the stimulus in the ROS 24/1 osteoblastic cell line [98]. It must be underscored that VDR involvement should be excluded because ROS 24/1 cells lack VDR expression. Similar studies on cultured chondrocytes derived from VDR knockout mice confirmed this observation [81]. Unfortunately, no receptor of the GPCR or receptor tyrosine kinase families dedicated to vitamin D or its metabolites has been identified so far. Nevertheless, the early studies of Nemere et al. on transcaltachia [99] resulted not only in the discovery of rapid response to vitamin D and its metabolites but also in the detection of a membrane-associated protein that binds to radiolabeled 1,25(OH)_2_D_3_ (KD-value 0.72 nM) [100]. The protein was initially described as 1,25D3-MARRS (membrane-associated rapid response to steroid), but it is also known as PDIA3 (protein disulfide isomerase family A member 3), GRP58, and ERp57 [101,102,103,104,105,106,107,108]. PDIA3 interacts with calreticulin (CALR) and calnexin (CANX) and plays a crucial role in the folding and export of proteins from the endoplasmic reticulum [80,109]. PDIA3 activity is essential for the proper function of the immune [74,110] and musculoskeletal systems [102,111] as well as for mammary gland growth and development [112]. Several groups have reported colocalization of PDIA3 with the cell membrane, cytoplasm, mitochondria, and even the nucleus [113,114]. Most importantly, PDIA3 was found to be involved in a rapid intestinal uptake of calcium and phosphate. Thus, its involvement in the classic function of 1,25(OH)_2_D_3_ was postulated [79,80,100,105,107]. It was also confirmed that 1,25(OH)_2_D_3_ binds to the a’ domain of PDIA3, with an estimated KD of 1 nM [115]. Site-directed mutagenesis studies revealed that the catalytic side of PDIA3 (C406) and its calreticulin (CALR) interaction site (K214 and R282) are required for the rapid response to 1,25(OH)_2_D_3_ [116]. Unfortunately, the nature of interaction of PDIA3 with vitamin D metabolites is not fully understood. The binding side for 1,25(OH)_2_D_3_ was not fully characterized because only a partial crystal structure of PDIA3 is available [117]. However, several studies have strongly supported involvement of PDIA3 in vitamin D signaling. For example, although a deletion of PDIA3 (PDIA3+/-) is lethal in mice [108,110], its partial silencing (PDIA3^+/-^) resulted in aberration in skeletal development [118], thus at least indirectly linking PDIA3 activity to calcium homoeostasis and vitamin D. Furthermore, targeted disruption of the PDIA3 gene results in an inhibition of rapid calcium transport and attenuates PKA or PKC signaling, thus impairing rapid nongenomic responses to 1,25(OH)_2_D_3_ [107,108]. In an addition, PDIA3 was also shown to interact with PLA2-activating protein (PLAA), which led to activation of phospholipaseA2 (PLA2), while targeted disruption of the PDIA3 gene attenuates cellular Ca^+2^ influx through L-type Ca^+2^ channels [119]. Other studies pointed out that uptake of calcium by the ER through the Ca^+2^ pump SERCA2b is regulated by PDIA3 and vitamin D [120]. It should be recalled, however, that the deletion of VDR also impairs bone formation [118], but it seems that PDIA3 but not VDR is essential for fast response to vitamin D, including activation of protein kinase C (PKC) signaling pathway [121] or a protection against UV-induced thymine dimer formation and DNA damage [122]. Furthermore, chondrocytes with a VDR knockdown but expressing PDIA3 showed a rapid increase in PKC levels and accelerated proteoglycan production in response to 1,25(OH)_2_D_3_. Thus, a distinct role for VDR and PDIA3 in 1,25(OH)_2_D_3_-mediated proliferation control of rat growth plate chondrocytes was postulated [81].

Association of PDIA3 with cell membrane signaling is strongly supported by observation of its interaction within caveolin-1 [123], which is the main scaffolding protein of caveolae (small invaginations of the plasma membrane which play important role in membrane singling and transmembrane trafficking). This observation at least partially explains the nature of PDIA3-driven rapid response to vitamin D through the association of PDIA3 with cell membrane caveolae and the activation of downstream signaling, including calcium mobilization, IP3, DAG, and cAMP production. PDIA3 was also found to regulate other intracellular pathways. For example, it was shown that 1,25(OH)_2_D_3_ triggers PDIA3-mediated rapid signaling cascade via CAMK2G (calcium/calmodulin dependent protein kinase II gamma), PLA2, PLC, and PKC, followed by an activation of mitogen-activated protein kinases (MAPK1 and MAPK2) [111,124]. Similarly, the PDIA3 membrane complex takes part in the rapid activation of WNT5A (Wnt family member 5A) by 1,25(OH)_2_D_3_-stimulated calcium influx and generation of secondary messengers, such as cAMP and/or IP3 [101]. Finally, some reports suggested nuclear localization of PDIA3 [114,125]. It is well established that PDIA3 has a noncanonic ER retention signal, Q/KEDL, but the presence of a Lys-rich nuclear localization signal was also suggested [111]. However, other studies suggested that TNFα, but not 1,25(OH)_2_D3, triggers fast translocation of PDIA3 to the nucleus [126]. In an addition, an active form of vitamin D was shown to trigger translocation of the complexes of PDIA3, with other transcription factors including STAT3 [127,128,129] and NF-κB [130]. Furthermore, binding of PDIA3 to DNA in close proximity to the STAT3 binding site was shown, and its regulatory role in the expression of at least a subset of STAT3-dependent genes was postulated [129]. Interestingly, other groups also showed that PDIA3 binds to specific fragments of DNA [104,131] and is involved in the regulation and expression of genes, including MSH6, TMEM126A, LRBA, and ETS1 [104]. Finally, PDIA3 was also found to support trafficking of retinoic acid receptor alpha (RARA) into the nucleus and subsequent degradation in Sertoli cells [132]. However, the effect of vitamin D on this process still requires verification.

In summary, the nature of direct or indirect interactions of 1,25(OH)_2_D_3_ with PDIA3 associated with membranes is still not fully understood since the biding side was not fully characterized. However, there is growing evidence that PDIA3 modulates the response to vitamin D. Furthermore, it is still under debate whether 1,25(OH)_2_D_3_ triggers translocation of PDIA3 into the nucleus and whether PDIA3 can act as a transcription factor or if PDIA3 only facilitates the trafficking of other transcription factors [133].

## 5. Direct or Indirect Regulation of Ion Transport by Vitamin D Metabolites

Keeping in mind that one of the best-described fast nongenomic targets of vitamin D is calcium influx, it could be speculated that vitamin D and its derivatives bind directly to ion channels or proteins affecting ion transport across the membranes. For example, the binding of sulfated and glucoronated derivatives of 25(OH)D_3_ to the multidrug resistance proteins SLCO1B1 (OATP1B1) and SLCO1B3 (OATP1B3) was described [134]. On the other hand, it is well established that 1,25(OH)_2_D_3_, through the genomic pathway, regulates expression of cell membrane transient receptor potential cation channels (TRPV1, 5, and 6), which at least in part explains the impact of vitamin D on calcium influx and cell proliferation [95,135,136,137]. Furthermore, 1,25(OH)_2_D_3_ increases expression of K^+^ two-pore domain channel subfamily K member 3 (KCNK3 expression) [138] but decreases the level of mRNA for potassium channels KCNH1 (Kv10.1) [139,140] and TASK-1 [138]. The electrophysiological studies on conductivity of ion channels showed that 1,25(OH)_2_D_3,_ at 100 nM concentration, acts as a mild agonist of the TRPV1 channel. In an addition, 25OHD and 1,25(OH)_2_D can also act as inhibitors of capsaicin-induced TRPV1 activity [136,141]. Molecular docking studies suggested that 1,25(OH)_2_D_3_ shared the capsaicin binding site at the vanilloid binding pocket of the TRPV1 [136]. Other studies suggested that the fast effect of 1,25(OH)_2_D_3_ on an ion transport may be mediated through activation of L-type calcium channels. Calcitriol triggered an increase in the L-type calcium current and the fast transient outward K^+^ current in myocytes. Furthermore, fast response to vitamin D depended on protein kinase A (PKA) and was absent in myocytes derived from mice with VDR knockout. Subsequent, intracellular Ca^2+^ mobilization resulted in ventricular myocytes’ contractility [142,143]. Finally, activation of chloride channels by vitamin D was found to be involved in the protection of human skin (ex vivo) from UV irradiation [144]. Interestingly, the modulation of mitochondrial potassium channels by 1,25(OH)_2_D_3_ has been shown recently [145] (the effects of vitamin D on mitochondria are discussed in Chapter 8). These intriguing electrophysiological observations require further investigations in order to provide more mechanistic details and the physiological significance of a direct impact of vitamin D on ion transport.

## 6. Nongenomic Activity of VDR

It must be underscored that the time frame (seconds to minutes) of rapid membrane responses to vitamin D virtually excludes the involvement of the transcriptional activity of VDR. Furthermore, multiple studies on the cell lines or model organisms with deletion of VDR have strongly suggested that rapid responses to vitamin D do not require the presence of vitamin D receptor (VDR). Thus, it seems that PDIA3, but not VDR, regulates the nongenomic activities of vitamin D [119,121,122]. However, the requirement of VDR in nongenomic activities of vitamin D may be strongly affected by the experimental model used in the study (e.g., cell type specific, type of assay). Furthermore, direct or indirect interaction between VDR and PDIA3 should also be considered. Indeed, several studies reported a membrane localization of VDR as well as colocalization with PDAI3 and caveolin 1 (CAV1) [146]. Interestingly, some reports have suggested that PDIA3 may serve as molecular chaperone for VDR [115,116]. However, as it was postulated above, at least some effects of 1,25(OH)_2_D_3_, such as rapid activation of PKC, are also observed in cells lacking VDR and thus are VDR-independent. On the other hand, a mutation within the isomerase catalytic side and removal of the KDEL motive (ER retention signal) of PDIA3 abolish this activity [116].

VDR was also detected in mitochondrial membranes [147,148,149] or even lipid droplets [150]. Interestingly, the existence of a second ligand-binding domain of VDR responsible for membrane signaling was postulated [27,151]. Although rapid activation of PLA2 was attributed to PDIA3, some other effects—such as 1,25(OH)_2_D_3_ activation of the SRC (SRC proto-oncogene, nonreceptor tyrosine kinase) WNT pathway [152,153,154,155], sonic hedgehog signaling molecule (SHH) [156,157,158,159,160], and NOTCH [161,162] signaling pathways [155]—were found to be VDR-dependent. Interestingly, VDR and NF-kB share DNA binding sites that may explain the influence of vitamin D on immune response [163]. Taken together, VDR may be involved in the activation of at least some of the nongenomic activities triggered by vitamin D, most probably in cooperation with PDIA3 [101,102,111,116,124,132,146,164,165].

## 7. Is Vitamin D as a Scavenger of Free Radicals or Their Source?

It is established that UVB is a major factor contributing to skin malignancy, but at the same time, it is essential for epidermal vitamin D production [60]. It could be hypothesized that during evolution, vitamin D may be adopted in direct or indirect response to radiation. Alternatively, UVB-driven vitamin D production may represent its primary function, which protected primitive aquatic organisms from UV or oxidative stress long before development of its endocrine function as a calcium–phosphate regulator.

The concept of a direct involvement of vitamin D in response to UV originated from a simple question: Can one overdose on vitamin D by sun tanning? This issue was addressed more than 40 years ago by Micheal Holick et al. in the classic paper [18]. It was shown that prolonged exposure to UVB did not lead to excessive accumulation of vitamin D. In fact, surplus of vitamin D was fast and efficiently removed by further structural rearrangements involving production of 5,6-transvitamin D_3_, suprasterols I and II in the skin [18]. Further studies revealed that irradiation of 5,7-dienes lead also to the formation of 5,7,9(11)-trienes with probable generation of a singlet oxygen, which may act as a photosensitizer [166,167]. Finally, it was shown that extensive irradiation of vitamin D and its analogs (such as isotachysterol) resulted in the formation of large groups of hydroxyl-, peroxy derivatives [18,168,169]. Detection of nonclassical products of UVB triggered photolysis of 5,7-dienes (7DHC and its natural or synthetic derivatives) might represent a natural mechanism of regulation of vitamin D levels as well as a cell protective mechanism. However, it is still not fully understood whether those generally unstable byproducts may be acceptors or also donors of reactive oxygen species (ROS). For example, accumulation of 5,7-dienes and 5,7,9(11)-trienes is important in the etiology of a 7Δ-reductase deficiency syndrome (Smith-Lemli-Opitz syndrome—SLOS) [166,167,170]. In this syndrome, the lack of the key enzyme converting 7DHC to cholesterol results in multiple metabolic defects with accumulation of 7DHC and its steroidal derivatives, including cholesta-5,7,9(11)-trien-3β-ol (9DDHC) [167] and pregna-5,7-diene-3b,17a,20-triol [170]. It was postulated that oxidation of 7DHC and 9DDHC leads to severe photosensitivity in SLOS patients with potential production of highly reactive singlet oxygen [166,171]. Nevertheless, it is still not known whether the potential phototoxicity of 7DHC derivatives is unique to SLOS patients or if it could be a general property.

Solar UV radiation reaching human skin damages cells and tissues directly or indirectly through the production of ROS, which are exemplified by superoxide and hydrogen peroxide [60,172,173]. UVB also induces DNA lesions, including formation of cyclobutane pyrimidine dimers (CPDs) [174], while UVB-generated ROS target guanine, leading to the subsequent generation of 8-oxo-7,8-dihydro-20-deoxyguanosine (8-OHdG) [55,175]. Unrepaired UV-induced DNA damage results in the accumulation of mutations, which contribute strongly to the formation of skin neoplasms [1,55,60]. On the other hand, it is well established that vitamin D protects skin cells from UVB-induced damage (including DNA damage and induction of inflammation), as was shown on various models, including in vitro, mice models, and human subjects [175,176,177,178]. Thus, it was postulated that cutaneous production of vitamin D protects skin cells from UV-driven carcinogenesis. The mechanism requires vitamin-D-induced upregulation of the genes involved in ROS response through VDR-mediated genomic pathways with additional activation of NRF2 [179]. However, PDIA3 was also shown to participate in cells’ protection against the formation of thymine dimers after irradiation with UV [122]. The effect was associated with activation of PKC signaling and calcium influx [180]. In addition, 1,25(OH)_2_D_3_ protected cells from UVR-induced thymine dimers via stimulation of chloride currents [144]. Photoprotective effects of 1,25(OH)_2_D_3_ were also associated with an increase in p53 tumor-suppressor protein translocation to the nucleus and a decrease in the level of nitric oxide species (NOS). Vitamin D can also affect endothelial function through the nongenomic pathway through the induction of PDIA3-mediated calcium, cAMP, Akt, and PKC downstream signaling, which in turn affect endothelial NO synthesis (eNOS) activity [181]. Finally, the potential UV-protective effect of 1,25(OH)_2_D_3_ and other vitamin D derivatives and analogues could be attributed to direct impact on mitochondria bioenergetics. It was postulated that the process of DNA repair induced by UV requires substantial energy expenditure, and 1,25(OH)_2_D_3_ and its analogs may directly affect mitochondrial activities (discussed below).

In summary, although vitamin D photoproduction may represent the primary and still-existing mechanism of cell protection against UVB and ROS/RNS, it must be underscored that vitamin D hormonal activity, namely VDR-mediated induction of the genes involved in UV and ROS/RNS, responses is also crucial for cell survival. On the other hand, potential involvement of 7DHC derivatives with triene moiety, such as 9DDHC in ROS generation (pro-oxidative), may be unique to SLOS.

## 8. Mitochondria as a Target for Vitamin D

Mitochondria are not only the powerhouse of the cell but also a key organelle to vitamin D metabolism, including activation, modification, and inactivation by the cytochromes of the P450 family (CYP27A1, CYP27B1, CYP24A1, and CYP11A1) [6,182,183]. All of those enzymes are located in the inner mitochondrial membrane. Thus, at least theoretically, all steps of the activation of vitamin D including hydroxylation in position 25 (CYP27A1) and position 1 (CYP27B1), as well as deactivation by hydroxylation at C-24, could be conducted within mitochondria. Although it must be acknowledged that CYP2R1 located in the endoplasmic reticulum is the major enzyme responsible for 25-hydoxylation, CYP27A1 also shows such activity [182,184]. Interestingly, CYP11A1 (CYP450scc—the major enzyme of steroidogenesis responsible for conversion of cholesterol to pregnenolone) was shown to also metabolize vitamin D, tachysterol, and lumisterol, which opens a new route of secosteroidogenesis [185,186,187]. There is no doubt that mitochondria are the direct target for vitamin D and its metabolites, but the mechanism of an entry, a regulation of vitamin D metabolism, or its direct impact on mitochondrial biogenesis or oxidative stress still requires an in-depth investigation. As was discussed previously, megalin may be responsible for vitamin D import to the cell. Interestingly, recent studies have suggested that megalin may also be involved in the translocation of some molecules directly to the mitochondria. Furthermore, mitochondrial megalin was found to be involved in the defense against ROS, and its knockout impairs mitochondrial respiration and glycolysis [93]. Megalin itself was colocalised with mitochondria in association with stanniocalcin-1 and SIRT3, which are involved in a defense against ROS [93]. Interestingly, it was shown that megalin is involved in the intracellular traffic and mitochondrial import of angiotensin II, stanniocalcin-1, and TGF-β [93], which raises the question of whether vitamin D and its metabolites, in concert with DBP, could be acquired and target mitochondria accordingly.

An active form of vitamin D also has indirect impact on mitochondria function through the VDR-dependent regulation of expression of genes involved in the oxidative stress response [145,183,188]. By using transcriptomic analyses focused on mitochondrial genes, it was discovered that 1,25(OH)_2_D_3_ inhibits expression of the several genes involved in oxidative phosphorylation and fusion/fission and upregulates genes involved in mitophagy and ROS defense [188]. In seems that the activation of selected genes’ coding proteins in response to ROS by 1,25(OH)_2_D_3_ is mediated by nuclear-factor-erythroid 2-related factor 2 (NRF2) [179,188]. NRF2 is a key transcription factor that can bind to antioxidant response elements (AREs) on DNA and initiate transcription of the genes responsible for the protection of cells against oxidative stress associated with diabetic neuropathy [188].

An involvement of VDR in mitochondrial physiology was confirmed by observation; deletion of VDR affects mitochondrial membrane potential, enhances ROS production [147], and results in subsequent mitochondrial damage [189]. Removal of VDR also resulted in an increased expression of elements of the respiratory chain, such as cyclooxygenase 2 and 4 (COX2 and 4), the ATP synthase subunits (6MT-ATP6 and ATP5B) [147], and subunits II and IV of cytochrome c oxidase [134]. Interestingly, VDR was also found in mitochondria, but in contrast to its nuclear translocation, its import to mitochondria is most probably ligand-independent [148]. VDR was found to interact with the mitochondrial permeability transition pore (PTP), the voltage-dependent anion-selective channel (VDAC). Recently, it was shown that vitamin D regulates the activity of mitochondrial calcium channels. Furthermore, VDR may also directly impact cholesterol and vitamin D transport through the interaction with the steroidogenic acute regulatory (StAR) protein [148]. The mitochondrial localization of VDR was linked to the redirection tricarboxylic acid cycle (TCA) towards biosynthesis, which is the common feature of neoplastic rather than fully differentiated cells [189]. Thus, it seems that VDR may affect mitochondrial function both through the activation of classic genomic pathways and also directly inside of mitochondria, including interaction with the mitochondrial genome. Interestingly, the protective anti-ROS properties of 1,25(OH)_2_D_3_ were shown on isolated mitochondria. Thus, the genomic effects of VDR activation may not be fully required [190]. It must also be underscored that 1,25(OH)_2_D_3_ may protect mitochondria from potential toxic insults, as it was shown that it can attenuate the cytotoxicity induced by aluminum phosphide via inhibiting mitochondrial dysfunction and oxidative stress in isolated rat cardiomyocytes [191].

## 9. Clinical Implications of Nongenomic Pathways Activated by Vitamin D

It is well established that 20 ng/mL of 25(OH)D_3_ in the serum is an adequate concentration for bone health. However, several studies and recommendations have suggested a concentration of vitamin D >30 ng/mL as optimal in order to provide extraskeletal benefits. Recently, it was postulated that even higher serum concentrations of vitamin D should be considered. For example, it was calculated that 37 ng/mL during pregnancy decreases the chance of complications in patients with risk of preeclampsia to values characteristic for the entire population [192]. Even higher concentrations (40–50 ng/mL) were proposed to be beneficial for extraskeletal outcomes of vitamin D, including its anticancer properties [193]. This leads to the significant change in recommendation concerning vitamin supplementation (800 IU vs. 4000 IU) and very high doses (50,000–100,000 IU) are recommended for patients with a severe vitamin D deficiency. However, potential hypercalcemia and hypercaluria must be considered as a potential side-effect of such as supplementation. However, several studies revealed that vitamin-D-induced hypercalcemia is observed usually in patients with defects in vitamin D metabolism (e.g., patients with *CYP24A1* mutation) [194]. This raises the very interesting question of whether an activation of the alternative nongenomic pathways is responsible for the additional phenotypical effects of vitamin D. It was shown that 1,25(OH)_2_D_3_ binds to VDR with an affinity of 0.1 nmol/L and to PDIA3 with an estimated Kd of 1 nmol/L [195]. Thus, it could be speculated that higher doses of vitamin D supplementation and higher (30 ng/mL or higher) serum concentrations of 25(OH)D_3_ are essential for an effective activation of nongenomic pathways. Nevertheless, this speculation requires in-depth laboratory and clinical investigations.

The description of transcaltachia was historically the first indication of existence of nongenomic pathways activated by 1,25(OH)_2_D_3_ [100]. This rapid vitamin-D-stimulated transport of calcium through intestinal endothelium is an important factor regulating the level of calcium in the body. Another clinical implication of nongenomic pathways is direct scavenging of UV-generated ROS and RNS observed as early as 15–30 min after treatment of irradiated keratinocytes with 1,25(OH)_2_D_3_ (please see [73] for discussion). In an addition to the clear UV-protective effects of vitamin D (see also chapter 7), it was postulated that vitamin D protects against skin photocarcinogenesis [57]. 1,25(OH)_2_D_3_ also affects ERK, PARP-1, and p53 activities and takes part in DNA protection and repair [73]. It is well established that increased production of ROS and deregulation of cellular energetics is a characteristic feature of cancer progression and drug resistance [196]. Thus, vitamin D may directly inhibit carcinogenesis through ROS scavenging. In fact, is also know that activation of VDR-dependent genomic pathway protects cells by activation of transcription of the genes involved in the response to ROS and DNA-repair [197]. Involvement of genomic pathways in the anticancer activities of vitamin D and its derives is also supported by observation that expression of VDR expression decreases with cancer progression [24]. However, it was also postulated that the extraskeletal benefits of vitamin D, including effects on cardiovascular and autoimmune diseases and cancer, are observed at relatively high concentrations [198]. Indeed, it was recently estimated that an increase in the serum concentration of 25(OH)D from 10 to 80 ng/mL would decrease cancer incidence rates by 70 ± 10% [60]. Thus, it is tempting to suggest that higher doses of vitamin D are required for the activation of alternative pathways, but this still remains to be confirmed.

Another, striking nongenomic activity of 1,25(OH)_2_D_3_ was shown on models of collagen or ADP-induced platelet aggregation. Interestingly, the inhibitory effect of 1,25(OH)_2_D_3_ on platelet aggregation varied depending on the state of diabetes [199]. Although VDR was detected in platelets and associated with mitochondria, because platelets naturally lack cell nuclei [149], only activation of nongenomic pathways should be considered. In an addition, it was also shown that glycemic control was inversely associated with high platelet aggregation and low levels of 25(OH)_2_D in diabetic patients [199]. These findings provide important clinical implication of alternative pathways activated by vitamin D.

The impact of PDIA3 and alternative pathways activated by vitamin D was also shown on an experimental model of cholangiopathy induced by Abcb4 knockout in mice. Firstly, additional silencing of VDR (VDR and Abcb4 double knockout) promotes a proinflammatory phenotype in cholangiocytes. Interestingly, vitamin D—or its analog, calcipotriol—treatment resulted in PDIA3-dependent reduction of inflammatory response [200]. Interestingly, PDIA3 was also found to be essential for the autolysosomal degradation of *Helicobacter pylori* [201]. Thus, activation of membrane response with potential involvement of PDIA3 may play a role in the regulation of immune response, including response against pathogens.

In melanoma, it was found that decreased expression of VDR correlates with poor prognosis. On the other hand, overexpression of PDIA3 was shown to be a poor survival factor in patients with clear cell renal cell carcinoma—ccRCC [127]. Thus, it seems that proper functioning of classic VDR-dependent pathways as well as activation of alternative pathways is essential in order to maintain body homeostasis as well as to protect from potential harmful insults, such as UVB or pathogen infection.

In summary, although genomic pathways activated by VDR still remain the major targets for vitamin D, with clear clinical implications (e.g., rickets), the existence of alternative pathways explains fast responses to vitamin D, and its direct impact on mitochondria and other intracellular processes. Unfortunately, the pleiotropic effects of vitamin D genomic activities, and the potential involvement of VDR in nongenomic regulation, hinder the studies of vitamin D membrane signaling, its effects on mitochondria, and the precise determination of the role of PDIA3 (Table 1).

## 10. Beyond 1,25(OH)_2_D_3_

Although it is not the major scope of this review, potential biological activities of vitamin D metabolites or analogs must be acknowledged. For years, it has been believed that calcitriol is the only active metabolite of vitamin D. In fact, the binding of 25(OH)D_3_ or 24,25(OH)_2_D_3_ to VDR is at least 100–1000 times weaker in comparison to 1,25(OH)_2_D_3_. Relatively recently, some alternative metabolic routes for vitamin D have been described, and some unique biological futures of those secosteroids have been proposed. In silico docking experiments showed the variable binding affinities to VDR of various alternative vitamin D metabolites or analogs. However, 1,25(OH)2D3 remained its main unquestioned ligand responsible for activation of the classic genomic response. It seems that alternative metabolites may alter or tune up or down transcription response to vitamin D. For example, 20(OH)D_3_ and 20,23(OH)_2_D_3_ generated by CYP450scc (CYP11A1) inhibit the activity of key factor in immune response NF-kB [202] or stimulate ROS response through NRF2 [179,187] similarly to 1,25(OH)2D3. However, it is believed that in addition to VDR, they may also target other transcription factors, such as RORα and RORγ [34,35] or AhR [28]. For many years, lumisterol and tachysterol we considered nonfunctional isomers of vitamin D, but recent studies have proposed their biological activity and further metabolism, which opens a new chapter in the field of secostroids [187,203]. Thus, it seems that 1,25(OH)_2_D_3_ is not the only functional metabolite of vitamin D and that other secosteroids may help in the fine-tuning of vitamin D response or even replace 1,25(OH)_2_D_3_, but it seems that higher concentrations are required.

## 11. Conclusions

Keeping the pluripotent properties of vitamin D in mind, the presence of alternative intracellular pathways may help to explain a variety of responses [204]. VDR genomic activity of course plays the central role in 1,25(OH)_2_D_3_ signaling. However, it must be noted that from the evolutionary perspective, some primal functions, such as protection from ultraviolet radiation, should be considered. On the other hand, membrane response, including calcium mobilization, may provide a fast route of action for this powerful secosteroid. The detailed mechanisms underlying membrane nongenomic responses still remain to be elucidated. However, the role of PDIA3, as well as that of membrane-associated VDR, should be considered. Furthermore, direct or indirect interaction of vitamin D metabolites with other proteins including megalin, ion channels, and as their impact on mitochondria may help us to understand the versatility of its phenotypic effects [64,186,205]. However, further in vivo studies and randomized controlled trials [206] are required to confirm physiological and clinical importance of alternative pathways of vitamin D signaling.

## Figures and Tables

**Figure 1 nutrients-14-05104-f001:**
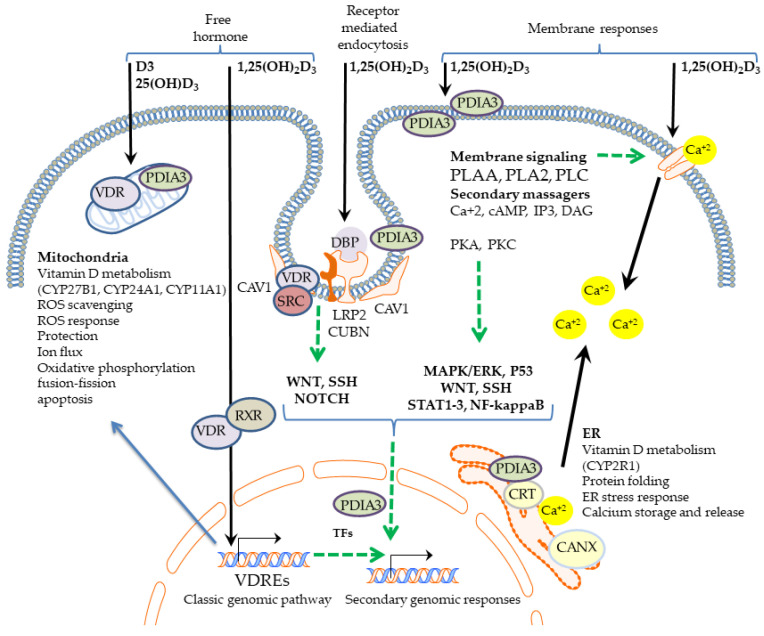
Vitamin-D-activated pathways. All secosteroids, including vitamin D_3_, 25(OH)D_3_, 1,25(OH)_2_D_3_, and other derivatives, enter the cell directly though the plasma membrane according to the free hormone hypothesis. Alternatively, 1,25(OH)_2_D_3_ bound to DBP may be imported by the megalin/cubulin complex (LRP2-CUBN complex) via receptor-mediated endocytosis. Inside the cell, 1,25(OH)_2_D_3_ binds the VDR-RXR complex, which is translocated into the nucleus and modulates the expression of targeted genes. Vitamin D metabolism takes place in the mitochondria, including activation of 25(OH)D_3_ by CYP27B1, resulting in formation of 1,25(OH)_2_D_3_ and inactivation of both metabolites by CYP24A1. In mitochondria, vitamin D may act as ROS scavengers, thus protecting the mitochondria from damage. Vitamin D metabolites may also directly influence ion transport, oxidative phosphorylation, and ROS response or may do so indirectly through the genomic pathway (blue arrow). PDIA3 mediates 1,25(OH)_2_D_3_-dependent membrane-signaling cascades. PDIA3 was detected on both sides of the cell membrane, on the endoplasmic reticulum, and in the cytoplasm, and its translocation to the nucleus was also postulated. PDIA3 mediates 1,25(OH)2D3-dependent activation of PLAA, PLA2, and PLC and the opening of Ca^+2^ and Pi (NaPi II a,c) channels. These result in the rapid accumulation of secondary messengers, including DAG, IP3, cAMP, and Ca^+2^, followed by PKA, PKC, or CAMK2G activation and the alteration of several downstream targets, including MAPK pathways. VDR was found also to be colocalized with CAV1 and SRC in caveolae. Its membrane activity was linked with the downregulation of WNT, SSH, and NOTCH signaling pathways. Alone or together with VDR after stimulation with 1,25(OH)_2_D_3_, PDIA3 modulates the transcription factors NF-kappaB, STAT3, and P53. At the endoplasmic reticulum, PDIA3—together with calreticulin (CRT) and calnexin (CANX)—facilitates protein folding. 1,25(OH)_2_D_3_, through VDR transcriptional activity, induces several transcription factors (TFs) which are involved in secondary genomic responses to this multipotent hormone.

**Table 1 nutrients-14-05104-t001:** Genomic and membrane pathways activated by vitamin D.

	Genomic	Membrane
Time course	Delayed response (hours–days)	Fast response (minutes–hours)
Primary Mechanisms	1,25(OH)_2_D_3_ binds to VDR, which is translocated to the nucleus together with RXR, where the complex binds vitamin-D-responding elements (VDRE) of DNA.	1,25(OH)_2_D_3_ affect activity of membrane proteins including PDIA3 and direct or indirect activity of ion channels. ROS scavenging.
Primary effects	Transcriptional regulation of up to 3000 genes resulting in inhibition of cell proliferation, induction of cell differentiation, immunomodulation, UVB response, ROS response, and alteration of mitochondrial function.	Membrane responses with activation of secondary messengers (calcium, cAMP, IP3, DAG). Transcaltachia, modulation of mitochondrial bioenergetics. Direct cell protection against UV, ROS, and pathogens.
Secondary mechanisms	Alteration of the expression of several TFs results in activation of secondary genomic responses.	Activation of secondary messages may activate MAPK/ERK, P53, WNT, SSH, STAT1-3, and NF-kappaB genomic activities. PDIA3 was found in the nucleus. Thus, its function as transcriptional regulator is considered.
Secondary effects	A dissection of primary from secondary genomic effects still requires careful investigation. Potential secondary impact of membrane signaling on genes expression adds an additional complexity to vitamin D response.	Modulation of activity of signaling pathways MAPK/ERK, P53, WNT, SSH, STAT1-3, and NF-kappaB results in modulation of cell physiology as well as activation of secondary genomic responses.
Role of VDR	VDR act as a transcription factor.	VDR may be engaged in membrane signaling.
Role of PDIA3	Potential secondary genomic effect is considered as PDIA3 was found in the nucleus.	PDIA3 is involved in initiation of membrane signaling.
Role in calcium regulation	Change in the expression of the genes responsible for calcium homeostasis	Rapid direct, or indirect effect on calcium transport (transcaltachia)
Effect on mitochondria	Indirect though alteration of the expression of mitochondria related genes	Potential direct effect on mitochondrial proteins (including cytochromes P450) and potassium ion channels from the inner mitochondrial membrane
UVB protection	Indirect through activation of stress response genes and DNA repair mechanisms	Potential direct ROS scavenging
Immunomodulation	Inhibition of B cell differentiation and immunoglobulin secretion, shift from a Th1 to a Th2 response, induction of T regulatory cells, decrease of expression of proinflammatory cytokines, and stimulation of expression of antimicrobial peptides.	Regulation of calcium influx may affect immune cell physiology.

## Data Availability

Not applicable.

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
