# Peer review of "Nongenomic Activities of Vitamin D"

_nutrients, 2022, doi:10.3390/nu14235104_

Round 1

Reviewer 1 Report

Following the analysis of the manuscript titled "Non-genomic activities of Vitamin D", I appreciate the article's topic is interesting and I recommend that it should be revised taking into account the following observations:

-          Abstract: please clarify the type of manuscript and what was the purpose of this article.

-    Please insert a table summarizing the main characteristics of the clinical studies describing the non-genomic activities of vitamin D.

-          I suggest adding a figure for vitamin's mechanism of action.

-      The manuscript should be based on presenting especially the latest evidence from the chosen topic. Therefore, update the references because too many articles in the list have been published for more than 10 years, even 20 years  (5, 10, 16-19, 26-28, 33, 67, 69-76, 78-81, 82, 84, 86, 91, 96, 97, 103, 111, 113, 122, 123, 126, 135-145, 149, 151, 163, 164, 167, 169, 170).

Author Response

I would like to cordially thank reviewer for helping me to improve the manuscript.

The manuscript has been corrected carefully according to reviewers’ suggestions.

1. Following the analysis of the manuscript titled "Non-genomic activities of Vitamin D", I appreciate the article's topic is interesting and I recommend that it should be revised taking into account the following observations:

-          Abstract: please clarify the type of manuscript and what was the purpose of this article.

Thank you for this remark:

Last sentence of the abstract has been modified:

“…, in this review their impact on cell physiology, as well as potential clinical applications will be discussed.”

2. -    Please insert a table summarizing the main characteristics of the clinical studies describing the non-genomic activities of vitamin D.

Thank you for this suggestion, because the second reviewer also requested to describe some clinically relevant non-genomic activities of vitamin D, I decided to add an extra chapter on the topic.

3. -          I suggest adding a figure for vitamin's mechanism of action.

Thank you for this remark. Please note that this is a review for special issue “Honor of Centennial of the Discovery of Vitamin D - The Central Role of Vitamin D in Physiology” and almost every manuscript, in this issue will have such a Figure. Please note, that Figure 1 which has been added to revised version of the manuscript actually describes classic pathway activated by vitamin D.

4. The figure summerising our current knowledge concerning none genomic pathways was included.

Thank you for this important comment, a dedicated Figure 1 was added:

5. -      The manuscript should be based on presenting especially the latest evidence from the chosen topic. Therefore, update the references because too many articles in the list have been published for more than 10 years, even 20 years (5, 10, 16-19, 26-28, 33, 67, 69-76, 78-81, 82, 84, 86, 91, 96, 97, 103, 111, 113, 122, 123, 126, 135-145, 149, 151, 163, 164, 167, 169, 170).

Thank you for this important comment, I fully agree with review, however having in mind the special issue: “Honor of Centennial of the Discovery of Vitamin D - The Central Role of Vitamin D in Physiology” I feel obliged to commemorate some classic (milstone) papers concerning vitamin D. Nevertheless, at least 20 citations 12 years or older have been removed or replaced by recent reviews.

Reviewer 2 Report

The paper reviews evidence suggestive of the presence of  non-genomic responses to vitamin D

Comment:

1.      The presentation is somewhat confusing. The author implicates genomics and VDR signaling while discussing the non-genomic effects leaving the reader somewhat confused. Can you include a table that lists genomic and non-genomic effects as well as effects that are difficult to assign to either one?

2.      Please discuss the clinical and therapeutic implications of the non-genomic vitamin D-triggered pathways, particularly via PDIA3.

3.      Is there really clinically significant difference in vit D beneficial effects between blood levels of 30ng/mL and 40-50g/ml? Cited references state that a supraphysiological dose of calcitriol is required to reduce the proliferation of cancer cells. However, the high dose will lead to calcemic side effects such as hypercalcemia and hypercalciuria. Can you cite a human control study showing clinically beneficial effects on cancer in humans? Also, are there data documenting that levels 40-50 ng/ml are associated with less incidence of cancer? Moreover, are there clear human data on anticancer effects that are not based on in vetro observation using isolated cancer cells?

4.      please discuss the effects on proinflammatory cytokine and microRNA.

5.      Please provide a figure to show both the genomic and non-genomic pathways that regulate the effects of vita D.

Author Response

I would like to cordially thank reviewer for helping me to improve the manuscript.

The manuscript has been corrected carefully according to reviewers’ suggestions.

The paper reviews evidence suggestive of the presence of non-genomic responses to vitamin D

Comment:

  1. The presentation is somewhat confusing. The author implicates genomics and VDR signaling while discussing the non-genomic effects leaving the reader somewhat confused. Can you include a table that lists genomic and non-genomic effects as well as effects that are difficult to assign to either one?

Thank you for this comment, requested table is included (Table 1). The main problem is that VDR it at least partially involved also in none-genomic activities of vitamin D, some non-genomic activities of vitamin D and in addition some none genomic pathway may also affect gene expression however not depended on VDR. In the revised version I have try to described complexity of intracellular pathways activated by vitamin D.

  1. Please discuss the clinical and therapeutic implications of the non-genomic vitamin D-triggered pathways, particularly via PDIA3.

Thank you for this important comment, a dedicated chapter was added:

“9. Clinical implications of non-genomic pathways activated by vitamin D”

  1. Is there really clinically significant difference in vit D beneficial effects between blood levels of 30ng/mL and 40-50g/ml? Cited references state that a supraphysiological dose of calcitriol is required to reduce the proliferation of cancer cells. However, the high dose will lead to calcemic side effects such as hypercalcemia and hypercalciuria. Can you cite a human control study showing clinically beneficial effects on cancer in humans? Also, are there data documenting that levels 40-50 ng/ml are associated with less incidence of cancer? Moreover, are there clear human data on anticancer effects that are not based on in vetro observation using isolated cancer cells?

The paragraph describing effects of vitamin D depending on it serum concentration, was added:

“It is well established that 20 ng/mL of 25(OH)D3 in the serum is adequate concentration for bone health, however several studies and recommendations have suggested concentration of vitamin D >30 ng/mL as optimal, in order to provide extra-skeletal benefits. Recently, it was postulated that even higher serum concentrations of vitamin D should be considered. For example, it was calculated that 37 ng/mL during pregnancy decrease chance of com-plication in patients with risk of preeclampsia to the values characteristic for the entire population.[165] Even higher concentrations (40-50 ng/mL) were proposed to be beneficial for extrasceletal outcomes of vitamin D, including its anticancer properties [166].].”

So far there are only a few randomized controlled trials on vitamin D supplementation and cancer.  Most of studies, ever recent focus on effect of supplementation rather then serum level of 25(OH)2D. The good example is VITAL trial, where actually two groups with similar vitamin D levels were compared. . In VITAL, the mean 25(OH)D concentration for those in the treatment group was near 30 ng/mL, for non supplemented around 25 ng/mL.

But according to recent Nutrients paper (Nutrients. 2022 Oct; 14(19): 4071):

The total mortality rate is about 50% greater among men whose 25(OH)D levels are lower than 46 nmol/L or higher than 98 nmol/L [60]; The risk of prostate cancer is significantly greater at levels lower than 40 nmol/L or higher than 60 nmol/L [61,62]; _ For endometrial, esophageal, gastric, kidney, non-Hodgkin’s lymphoma, pancreatic,  and ovarian cancer, the mortality rate is significantly greater at levels lower than 45 nmol/L or higher than 124 nmol/L [63];  _ The risk of pancreatic cancer is significantly greater at levels higher than 100 nmol/L [64];  _ The risk of cardiovascular disease is significantly greater at levels lower than 50 nmol/L or higher than 62.5 nmol/L, and the mortality rate for all causes is significantly greater at levels higher than 122.5 nmol/L [65].”

I think presented information clarifies our current understanding on the impact of vitamin D on cancer.

  1. please discuss the effects on proinflammatory cytokine and microRNA.

The effects of vitamin D on immune response is not the main topic of the manuscript, however it was mentioned in several points. A new paragraph concerning of potential clinical implication was added (please see new chapter 9). Several recent papers investigated role of vitamin D in regulation of microRNA expression, but as far as I know all papers focused on genomic activity of vitamin D driven by VDR, thus are out of the scope of this review (e.g. J Nutr Biochem. 2022 Nov;109:109105 or Transl Cancer Res 2021 Jun;10(6):3111-3127.) If the reviewer is willing to provide some clues (paper to cite) I would be happy to include them. In order to acknowledge the impact of vitamin D on miRNA, the sentence was added:

“Recently, regulatory effects of 1,25(OH)2D3 on the expression of microRNAs and long non-coding RNAs has been uncovered as a potential anti-cancer mechanism.”

  1. Please provide a figure to show both the genomic and non-genomic pathways that regulate the effects of vita D.

Thank you for this important comment, a dedicated Figure 1 was included.